



# Design and implementation of a mobile device APP for network-based EEW systems: application to PRESTo EEWS in Southern Italy

Simona Colombelli[1], Francesco Carotenuto[1], Luca Elia[1] and Aldo Zollo[1]

[1]Department of Physics "Ettore Pancini", University of Naples Federico II

*Correspondence to*: Simona Colombelli (simona.colombelli@unina.it)

**Abstract.**

A fundamental feature of any Earthquake Early Warning System is the ability of rapidly broadcast earthquake information to a wide audience of potential end users and stakeholders, in an intuitive, customizable way. Smartphones and other mobile devices are nowadays continuously connected to the internet and represent the ideal tools for earthquake alerts dissemination, to inform a large number of users about the potential damaging shaking of an impending earthquake.

Here we present a mobile App (named ISNet EWApp) for Android devices which can receive the alerts generated by a network-
based Early Warning system. Specifically, the app receives the earthquake alerts generated by the PRESTo EWS, which is currently running on the accelerometric stations of the Irpinia Seismic Network (ISNet) in Southern Italy. In the absence of alerts, the EWApp displays the standard bulletin of seismic events occurred within the network. In the event of a relevant earthquake, instead, the app has a dedicated module to predict the expected ground shaking intensity and the available lead-time at the user position and to provide customized messages to inform the user about the proper reaction during the alert. We
first present the architecture of both network-based system and EWApp, and then and describe its essential operational modes. The app is designed in a way that is easily exportable to any other network-based early warning system.





## 1 Introduction

When an earthquake occurs the rapid assessment of its impact is essential for timely and appropriate emergency operations, such as securing people and crucial infrastructures exposed to serious damage effects. Earthquake Early Warning Systems (EEWSs) are now starting to be considered as an effective strategy for the real-time earthquake risk reduction. EEWS are real-time modern information systems aimed at providing rapid notification of the potential damaging effects of an impending earthquake, through the rapid telemetry and processing of data from dense instrument arrays deployed in the source region of

the event of concern or surrounding the target infrastructure. A crucial feature of any EEWS is the ability of communicating rapid earthquake information to potential end users and stakeholders in a user-friendly, user-oriented, and customizable way. With the increasing, huge, worldwide spread of mobile cell-phone technologies and wireless internet connection, smartphones are the ideal candidates for receiving broadcast alerts.

Several operational worldwide EEWS are now releasing earthquake alerts leveraging smartphone technologies, with dedicated

applications which may act as broadcasting target or may even transform the smartphone into a seismic detector. Among them, Japan is the leading country for EEW development. The current Japanese EEW system, operated by the Japan Meteorological Agency (JMA, https://www.jma.go.jp/jma/indexe.html) has several broadcasting channels, including TV, radio, Internet, dedicated EEW-capable devices and cell-phones devices. Since October 2007, the three major mobile phone companies have developed a broadcasting system to send multiple users an SMS of the EEW. Also, it is mandatory for 3G cell phones, sold

after 2007, to receive the warnings issued by JMA and release EEW notifications.

Another example of the use of smartphones for EEW is the MyShake App (Kong et al., 2016), which was recently developed by the EEW research group at the University of California at Berkeley. MyShake App has the ability to recognize earthquake shaking from background noise using the sensors in every smartphone. When a potential earthquake is detected by a single smartphone, its app sends the collected data to a centralized processing hub. Aggregating the information of the multiple

recording devices in the nearby area, a network detection algorithm may confirm or cancel the presence of an ongoing earthquake and possibly provide its location and magnitude. Source parameters are then used to estimate the shaking intensity and the available lead-time at any target location.

After the 2011 Tohoku earthquake in Japan, Taiwan also started to develop the Cell Broadcast Services (CBS), for disclosing emergency alerts to the public. Since May 2016, the Central Weather Bureau (CWB) is the main central agency for issuing

earthquake disaster alerts in Taiwan for events with magnitude greater than 4.5 occurring in and around Taiwan. The Central Weather Bureau further developed four apps (in Chinese) which are released for free by private developers, in cooperation with CWB. Users can receive real-time notifications of the location and magnitude of an earthquake and the earthquake's intensity in different areas (S.-M. Chang, 2018)



In Mexico, the official earthquake alerting system is run by the government-funded no-profit agency CIRES
(http://www.cires.org.mx/index_in.php), which was created after the Michoacan earthquake in 1985 that killed thousands of
people in the country. Based on about 100 sensors, mostly deployed along the Pacific coast of Mexico, CIRES transmits
earthquake early warning alerts through a network of VHF stations and offers alert systems for buildings and personal use.
More than 90,000 users in Mexico City, including almost all public schools, have CIRES receivers. The Mexico City Metro
additionally receives SASMEX alerts (Suárez et al., 2018), although not for public dissemination but instead to stop trains or
delay departures as necessary. More recently, in 2013, a private company, Sky Alert, former partner of CIRES, developed its
own earthquake early warning system which warns people through a mobile App (https://skyalert.mx/). When an earthquake
is detected by the closest SkyAlert sensor of a nation-wide seismic network, the recoded shaking intensity is evaluated and
confirmed by the nearby recorders and a broadband alert signal is sent to the control center of SkyAlert. The alert is then shared
through the MS cloud Azure, that broadcasts the message to all the SkyAlert App users, triggering a loud sound, a vocal
message of seismic alert and a text message with the information about the size of the earthquake. Since the two large
earthquakes that hit Mexico in September 2017, SkyAlert has doubled its users to 5.8 million, making it one of the country's
most downloaded apps.

In Europe, several cell-phone applications have been developed by local and national agencies (for example INGV,
http://www.ingv.it/it/; EMSC, https://www.emsc-csem.org/), with the major purpose of releasing earthquake information to
the massive audience of smartphone users, in the minutes following a relevant seismic event. Most of these apps operate as a
standard seismic bulletin, by collecting and reporting the earthquake information as released by the reference agency. Common
additional features are generally implemented, such as the possibility to activate customized notifications, in case of a
significant earthquake occurred nearby the selected target location or, for the largest events, occurred at any distance from the
user. Another relevant feature of these apps is the possibility for the users to provide their feedback after experiencing an
earthquake, such as the level of shaking perceived. This complements the traditional methods of collecting data (i.e., using
traditional seismic instrumentation) with information easily obtained through the internet, for an improved mapping of the
damage area, in the minutes after a seismic event.

The Earthquake Network project (Finazzi, 2016) started in 2013 with the aim to network cellular phone users living in active
seismic areas in order to provide a rapid earthquake alert based on the strong shaking recorded by the built-in accelerometric
sensor. The project has a similar objective to that of the Quake Catcher Network project (Cochran et al., 2009) and the
Community Seismic Network monitoring system (Clayton et al., 2011). When the smartphone detects a vibration, it sends a
signal to a central server network with information about the location of the smartphone. The central server checks whether
the number of received signals is anomalous, high relative to active smartphones located in the same area, and in that case an
earthquake warning is issued to the network.

Within this context, here we developed a mobile app for Android devices (smartphones, tablets and smartwatches) to receive
the broadcast alerts issued by PRESTo (Satriano et al., 2010), which is the EEWS currently operative in Southern Italy, aimed
at detecting small-to-moderate earthquakes occurring in the Campania-Lucania Apennine region. The mobile app has the main



purpose of informing a wider community of smartphone users about the incoming arrival of ground shaking due to earthquakes occurring in the target region, which represents one of the highest seismic risk areas of the country. Moreover, in the absence

of earthquakes, as it is done by existing, similar seismological apps, the standard bulletin of the events detected by the network, with all the available earthquake source information, is provided. We first synthetically describe the network infrastructure and PRESTo EEWS, and then discuss the functionality, the scientific methodologies and the architecture of the EWApp.

## 2 Scientific background and infrastructures: the Irpinia Seismic Network and PRESTo EEWS

The EWApp described here has been originally conceived to be interfaced with any regional, network-based EW system and

it can receive and elaborate standard output parameters from various EW platforms, such as PRESTo (http://www.prestoews.org/), Virtual Seismologist (Cua and Heaton, 2007), ElarmS (Allen R.M., 2007). Here we focus and describe the interaction of the EW app with the ISNet network and PRESTo EW system.

The Irpinia Seismic Network (ISNet, Weber et al., 2007; Iannaccone et al., 2009) (Figure 1) is a dense, local network of strong motion, short period and broad band seismic stations deployed along the southern Apenninic chain covering the seismogenic

areas of the Irpinia region. The area is one of the highest seismic risk regions of the Italian territory and is the location of several moderate-to-large earthquakes that occurred in the last centuries, including the Ms=6.9, 23 November 1980 event, and the 1996 (M 5.1), 1991 (M 5.1) and 1990 (M 5.4) events. The seismic network is composed of 30, real-time stations which continuously monitors the background seismicity and transmit the recorded signals to a dedicated control center. The stations are organized in sub-nets, each of them is composed of a maximum of 6-7 stations. The stations of a given sub-net are connected

with real-time communications to a central data-collector or Local Control Center (LCC). The different LCCs are linked among them and to a Network Control Center (NCC) with different type of transmission systems. The whole data transmission system is fully digital over TCP/IP, from the data-loggers, through the LCC, to the NCC, located in the city of Naples, 100 km away from the network center. To ensure a high dynamic recording range, each seismic station is equipped with a strong-motion accelerometer and a three components velocity meter (natural period=1 sec). In five station locations the seismometers are

replaced by broad-band (0.025-50 Hz) sensors to guarantee the well recording of teleseismic events.

The EEWS PRESTo is currently operative in the Campania-Lucania Apennine region, to rapidly detect and characterize the small-to-moderate earthquakes occurring in the area. PRESTo (PRobabilistic and Evolutionary early warning SysTem; Satriano et al., 2010) is a software platform for EEW that integrates algorithms for real-time earthquake location, magnitude estimation and damage assessment into a highly configurable and easily portable package. PRESTo continuously processes

the live streams of 3-components acceleration data from the stations for P-waves arrival detection and, while an earthquake is occurring, it promptly performs the event detection, location, magnitude estimation, damage zone assessment and peak ground-motion prediction at pre-defined target sites (see Method section for details) (Figure 2).

Our EWApp is able to further elaborate the alerts generated by the backbone infrastructure and PRESTo, in order to account for the current geographical position of the user and to calculate the amplitude and arrival time of the damaging seismic waves.



The EWApp, developed in Java, is the client of a client-server system called *EWAppSystem*. Once installed, the EWApp continuously runs in background mode on the Android devices and starts releasing alert notifications as soon as an earthquake is detected by PRESTo. During an earthquake, a key feature of the EWApp is a specific module to predict the expected level of ground shaking at the target location, within a maximum distance of 200 km from the epicenter position. In this way, the potential area of interest for the EWApp users covers both the Irpinia seismic region and the surrounding area.

## 3 The EWApp: methodologies and graphical user interface


The EWApp has a double operation mode: it can operate in *passive mode* (as a seismic bulletin) and in *active mode*, as a warning device. The block diagram of Figure 3 shows an overview of the whole system with an illustration of the main steps and links of the app, while the specific features of each operation mode are described in detail in the following dedicated sections. The app is available in two languages (English and Italian) and when installed, the current language of the running operating system is automatically selected.


### 3.1 Passive Mode

The first mode (*passive mode*), is similar to a standard seismic bulletin and allows for the visualization of seismic events that can be of interest for the smartphone user. When operating in the passive mode, the app duplicates the list of the events recorded by the ISNet network occurred at a distance smaller than 200km from the user's position. The events are included in the app

bulletin after they have been manually revised by the operator (typically, the day after the occurrence). The earthquakes can be sorted by choosing among three available options: by date (origin time), distance from the current position of the user or magnitude (Figure 4a). As the user taps on an event in the list, the relevant features of each earthquake are available to the user. Specifically, for each earthquake, the EWApp shows the current user position and the epicenter on a map, together with: origin time (both local and UTC time), local magnitude, epicentral coordinates (latitude and longitude), hypocenter depth and

distance from the current user position (Figure 4b).

### 3.2 Active Mode

When an earthquake is detected by PRESTo, the app starts working in the *active mode*. The input data are the evolutionary estimates of source parameters, as released by PRESTo and consisting in the estimates of the earthquake location (hypocentral

coordinates and origin time) and magnitude. Specifically, the earthquake location uses an advanced, evolutionary real-time technique based on an Equal Differential Time (EDT) formulation, and a probabilistic approach for defining the hypocenter (RTLoc, Satriano et al., 2008), using both the information from triggered arrivals and not-yet-triggered stations. Magnitude estimation is based on the evolutionary measurement of peak ground-displacements amplitudes, measured over the first $2 \div 4$ seconds of signal starting at the detected P-wave arrival and the estimated S-wave arrival and on the use of an empirical



relationship that correlates the final event magnitude with the logarithm of the initial peak amplitude (RTMag, Zollo et al., 2006).

As soon as the first output estimates are released by PRESTo, the app receives these input data and starts computing its own further estimates. The theoretical arrival time of the S-wave are first computed at the user position, by assuming a homogeneous velocity model for the wave propagation (vs=3.3 km/s). Then, using the estimates of location and magnitude, and a standard

Ground Motion Prediction Equation (GMPE) (Bindi et al., 2011) the expected level of ground shaking, in terms of Peak Ground Motion (PGV), is computed at the user position. Depending on the distance from the user and on the predicted PGV value, the APP can decide whether to issue the warning or not.

In case of a faraway earthquake (i.e., hypocenter $\geq$ 200km from the user position), independently of the expected intensity, a push notification (with default sound and vibration) warns the user that a seismic event has occurred within ISNet, although

its impact at the user site is negligible (Figure 5a). When tapping on the push notification, the relevant event parameters are available in a dedicated list of received PRESTo alerts (in the same format as those available in the standard ISNet bulletin).

In case of a closer earthquake (i.e., hypocenter < 200km from the user position), depending on the comparison between the expected intensity ($I_{MM}^{pred}$) and the threshold intensity value ($I_{MM}*$), two situations are possible: *no alert* and *alert*. For the specific application to the Irpinia seismic region, the threshold level for the alert declaration is currently set to $I_{MM}$=4, which

corresponds to a level of perceived shaking, based on the PGV-to-$I_{MM}$ conversion table from Faenza and Michelini (2010).

### $I_{MM}^{pred} < I_{MM}*$: no alert

If the predicted intensity does not exceed the threshold value, a simple push notification appears on the display, with information about the hypocenter position and expected shaking level (negligible or weak) (Fig 5b). As in the previous case, when tapping on the push notification, the bulletin information is shown (list of received alerts).

### $I_{MM}^{pred} \geq I_{MM}*$: alert

When the predicted intensity exceeds the threshold level, the alert mode is activated. From this moment ahead, every second, depending on the earthquake location and relative distance to the user position, the display shows the countdown with the available lead-time (e.g. the time available for safety actions before the arrival of strong shaking waves) and the predicted level of intensity. The lead-time is computed as the remaining seconds before the arrival of S-waves and is updated every second.

To avoid jumps and discontinuities due to small changes in the earthquake location, the countdown is updated and re-started only when the real-time, current lead-time value differs from the previous one more than 5 seconds. As for the alert levels, these are progressively updated following the output in terms of location and magnitude released by PRESTo.

In the event of a damaging earthquake, the smartphone display shows a map with the epicentral position of the event, the position of the user at the current time and the instructions to react. The instructions are given in the format of "Drop! Cover!

Hold On!" and are repeated, both in the form of images and voice messages, until the arrival of the S-waves (Figure 6a,b).

A fundamental, innovative feature of the EWApp is the identification of the end of the event, to warn the smartphone users that the strongest shaking has passed, and the emergency time is finished. To this purpose, the app includes an ad-hoc module that theoretically computes the expected end-of-shaking, based on the estimated magnitude of the event. The duration of the


shaking ($\tau_{shake}$) is defined as the time interval between the arrival of the P-wave and the moment at which, after reaching the

peak, the ground velocity decreases back down to a predefined threshold value. The threshold value is set to 0.2 cm/s, which is the lower threshold for the level of Macroseismic Intensity equal to 4, based on the PGV intensity regression of Faenza and Michelini, (2010), and is associated to a perceived level of "light" shaking. For each earthquake, the duration of the shaking ($\tau_{shake}$) is computed as:

$$\log (\tau_{shake}) = a+bM, \qquad\qquad\qquad (1)$$

where M is the estimated magnitude of the ongoing event (as provided by PRESTo) and a and b are empirically derived coefficients. The coefficients have been established based on the analysis of a large dataset of earthquake records from Italian and Japanese earthquakes, in the magnitude range between 3.5 and 9, and in the distance range from 0 to 200 km, which is the regional distance range the app is targeted to, for a total of 4036, 3-component records. For each available record, we measure the shaking duration ($\tau_{shake}$) on the horizontal components of the ground velocity (as vector composition) and determine its

scaling with magnitude, after verifying that the shaking duration is pretty insensitive to the distance, in the considered distance range. The Supplemental Material contains the full theoretical explanation and the details of the computation of the shaking duration. We found a=-0.58 and b=0.35. (Figure 7).

Finally, when the alert expires, the user has the possibility to notify his own health condition, and to communicate it to a list of pre-defined contacts. To this purpose, two intuitive buttons are positioned on the screen to communicate a safety state (green

button, "*I am fine*") or to ask for a help (red button, "*I need help*") (Figure 6c). In both cases, the EWApp obtains the current geographical position and sends a standardized text message to the list of contacts, including position and condition of the user. The contact list can be created when the EWApp is installed on the smartphone for the first time and can be changed later using a dedicated functionality within EWApp. Finally, the standardized text message can be also shared through the most common social network platforms, such as Facebook and Twitter, if the user personal account is available.

**4 Early Warning App Implementation and System Architecture**

PRESTo sends real-time alert messages as soon as an earthquake is detected and its source parameters (e.g. location, origin time and magnitude) are estimated. A new message is also sent as soon as those estimate change (they improve with time as new information is available), thus making the last message the most authoritative. Each message is encoded in a standard and flexible format used in seismology, QuakeML (http://www.quakeml.org). The QuakeML message is sent to a message broker

(such as ActiveMQ, http://activemq.apache.org/) using the STOMP protocol (https://stomp.github.io/). The message broker server is then able to broadcast the message to a large number of connected clients.

In practice, however, mobile devices are usually configured to not maintain a permanent connection to the Internet in order to avoid consuming excessively the battery. Moreover, they are not able to process and display in real-time a set of alert messages sent within a few tens of milliseconds, like those that can be generated by PRESTo during the processing of a single earthquake.

For these reasons, to make sure that the smartphones receive in real-time the alert messages must be sent through a cloud



messaging service such as Fire Cloud Messaging (FCM) (Figure 8). The alerts sent via FCM can in fact awaken the device even when it is in standby, thus starting the process illustrated in the previous paragraph.

To reduce the number of broadcast messages (in order to avoid excessive network traffic and improve scalability) it is necessary to apply a filter that selects the messages to be sent based on their relevance (e.g. maximum one message every second, the
magnitude or the hypocentral distance between two consecutive sent messages must vary appreciably, etc.). To filter the incoming messages and to send them to FCM, a server proxy component called Middleware-EWApp (MEWApp) has been implemented. MEWApp is a Python software that processes the incoming messages received from PRESTo via ActiveMQ, applies the above filtering criteria, extracts the most relevant information (location, origin time and magnitude) and formats them into an FCM message (JSON) and sends them to the FCM cloud service. FCM is then finally responsible for forwarding
these messages to all the installed EWApps which carry out the computational scheme described above, such as the shaking calculation based on the user position.

## 5 Discussion and Conclusions

We developed an intuitive and user-friendly app for Android mobile devices, which is able to receive and elaborate standard output parameters from a network-based EW platform and perform earthquake alert notifications, in the event of a relevant
damaging earthquake. The EWApp is highly flexible and, in principle, it can interact with various EW platforms (such as PRESTo, VS, Elarms), as long as the output parameters are provided in a standardized format (QuakeML). Whichever the EW platform is, the EWApp developed here is conceived to be a broadcasting channel of earthquake alerts and requires a backbone infrastructure for data collection, streaming and analysis. Here we tested the EWApp and its interface with ISNet network and PRESTo EW software. Specifically, we distributed the app to a limited number of academic users (within RISSC-Lab group
and the Department of Physics, "E. Pancini") to verify the basic functionalities and receive feedback from the users.

Consistently with what is done by similar apps, once an earthquake is detected (by PRESTo EWS, in our case), the EW app is able to geolocate the user position and use it to decide in real-time how to operate, i.e., whether to issue the warning or not. A relevant feature of the app is indeed an intelligent Decision Module which receives the standard output parameters from the core EW infrastructure, elaborates them and computes the expected intensity at the user position. Depending on the distance
from the event and on the expected damage, the EWApp activates personalized alert messages, containing the available lead-time, the predicted level of intensity, as well as the instructions for the behavior during emergency times. The estimates are continuously updated and refined with the passing of time, and the warning mode keeps being active as long as the ground shaking is ongoing.

The additional, most innovative feature of the EWApp is the double communication way, which allows receiving as input the
real-time earthquake parameters from the EW core platform and communicating (as output, with a dedicated module) the state of health and condition of the user, at the conclusion of the emergency time. This functionality is of extreme relevance



especially in the context of densely populated areas (such as urban areas, big industrial settlements), to collect the condition of people at the end of the event, for an efficient planning of the rescue operations.

In a perspective new release of the EWApp, several features could be implemented with special regard to the user's health

monitoring. For example, the app could be interfaced with a decision/control expert system, combining the outputs from the regional EEW with information from local monitoring devices (sensors), for broadcasting personalized safety instructions and customized alert messages, depending, for example, on the location of the personnel inside a given area (or inside a building). Moreover, the current geolocation functionality could be further improved, by adding the possibility to track the position and condition of people before, during and after the earthquake occurrence. This could be done using the user device, for example,

by coupling accelerometric data recorded by the smartphone to monitor the user's movements with data related to the operation and use of the device itself (i.e., web accesses, active calls, outgoing messages…). Additionally, the position and condition of the user as monitored by the smartphone could be coupled with some other health status parameters, as provided by different devices, such as heartbeat/pressure sensors. In both cases, intelligent Neural Networks algorithms, specifically trained, can be used to identify a condition of inactivity and quiescence, which could be synonymous of users in dangerous conditions or

potentially trapped under the earthquake ruins.

The EWApp is currently conceived to receive source parameter estimates from a network-based EW platform. An additional, parallel module could be easily included in the app to receive ground shaking parameters from a single station, on-site, independent EW platform. In this case, the EWApp would directly receive from the EW platform simplified pieces of information, in the form of alert levels, quantifying the occurrence of a large/small earthquake, distant/close to the user position

(Zollo et al., 2010).

Finally, another perspective idea to be implemented in a new release of the app would be to include a "drill-test mode", that is the possibility of running playback and scenario earthquakes from a dedicated list of events, in order to test, for example, the procedures for the evacuation of people from buildings and or densely populated areas. To this purpose, an upgraded version of the app could be specifically developed and released to advanced users only (such as private/public stakeholders and end-

users). This version could be linked to a dedicated software, allowing to activate the playback mode and to collect confidential data related to the users position and reaction to the drill, which could be in turn analyzed by expert social scientists, to delineate the profile of the community involved in the drill.

**Supplementary Material** is linked to the online version of this paper.

**Author contribution:** S.C. wrote the manuscript, performed the analysis of data and prepared the figures; F.C. developed and

implemented the EWapp; F.C. and L.E. built the link between the EWApp and the backbone infrastructure; A.Z. contributed



to the manuscript conception, design, and preparation. All authors equally contributed to the scientific discussions for the implementation of the app and to the manuscript revision.

**Competing interests:** the authors declare no competing interests.

**Acknowledgements:** the work was financially supported by the University of Naples, Federico II. Active faults for the Irpinia
region are taken from Database of Individual Seismogenic Sources (DISS), available at http://diss.rm.ingv.it/diss/. Data for Italian earthquakes are made freely available by the ITalian ACcelerometric Archive, ITACA 2.0 (http://itaca.mi.ingv.it). We acknowledge the Japanese National Research Institute for Earth Science and Disaster Prevention (NIED) for making the Kik-Net and K-net strong motion Japanese data accessible through website (http://www.kyoshin.bosai.go.jp/). Most of the analysis and figures were made using the GNUPLOT (http://www.gnuplot.info/, last accessed August 2019), the Seismic Analysis
Code (Goldstein et al., 2003; http://www.iris.edu , last accessed August 2019), the Generic Mapping Tools software (Wessel and Smith, 1995; https://www.soest.hawaii.edu/gmt/, last accessed August 2019) and OpenStreetMaps (© OpenStreetMap contributors; https://www.openstreetmap.org/copyright/en, last accesses August 2019).

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

355



**Figure 2: Scheme of PRESTo operations and IN/OUT connections.** The figure shows a simplified scheme of PRESTo, illustrating the steps of the software which are mostly relevant to the EWApp (i.e., event detection, event location and magnitude estimation). Three output channels are currently available for PRESTo, while the EWApp represents an additional output mode (through the ActiveMQ message broker).





**Figure 3: Block Diagram of the EWApp.** The figure shows the block diagram of the EWApp, with its operation modes. The left side shows the passive mode, in which no relevant event is detected from PRESTo and the App works as a standard seismic bulletin. The right side of the block diagram illustrates the various steps and operations during the active mode, both in case of a distant event and in case of a close event.


**Figure 4: Passive Operation Mode and earthquake bulletin.** The figure shows screenshot examples of the app when working in passive mode. Panel a) shows the earthquake bulletin with its main functions (sorting, bug reports). Panel b) shows the details of the event, appearing when tapping on a specific earthquake. The map was created using OpenStreetMaps (© OpenStreetMap contributors 2019. Distributed under a Creative Commons BY-SA License.).





Figure 5: Active Operation Mode with no alert. The figure shows screenshot examples of the app when working in active mode with no alert activation. Panel a) shows the case of a distant event, while panel b) shows the case of a closer event for which the expected shaking does not exceed the threshold. In both panels, the top image shows the pop-up notification appearing on the screen at the occurrence of the event. The maps were created using OpenStreetMaps (© OpenStreetMap contributors 2019. Distributed under a Creative Commons BY-SA License.).



385

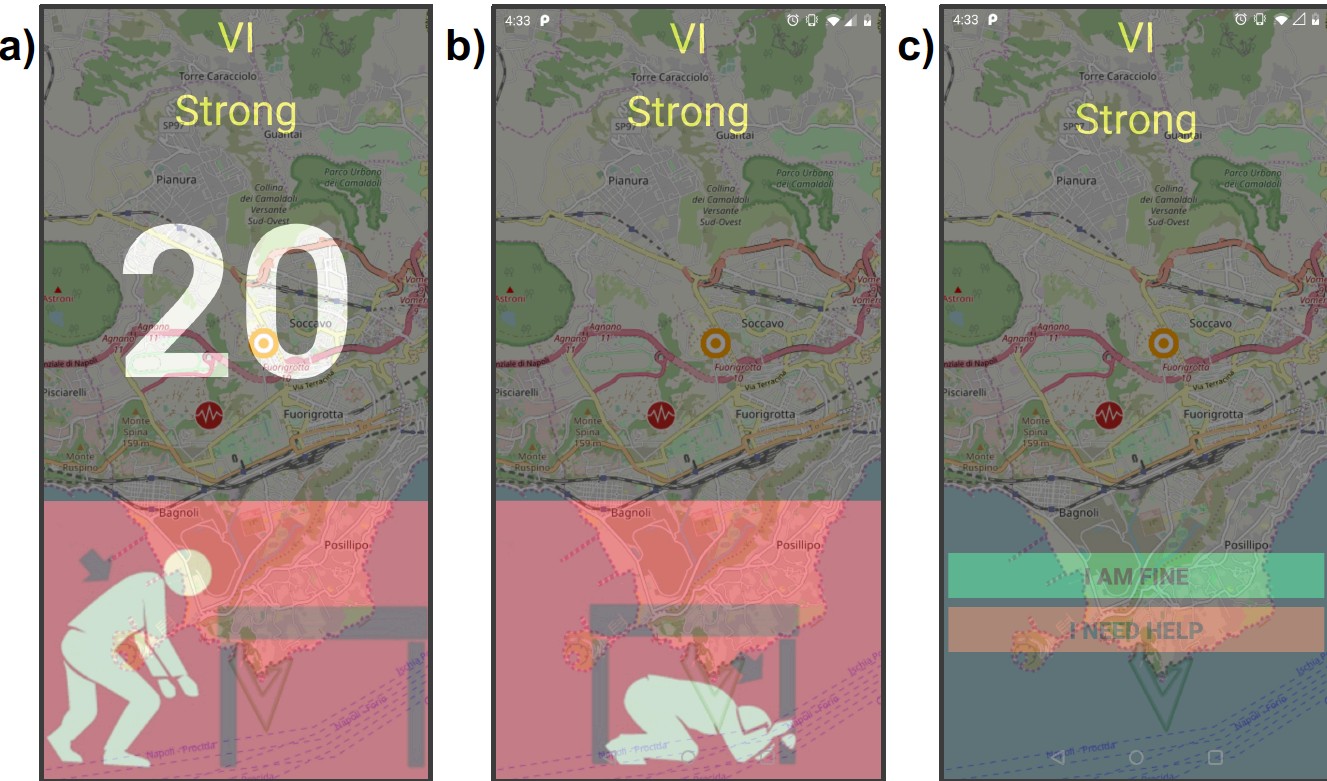

**Figure 6: Active Operation Mode during an alert.** The figure shows screenshot examples of the app when working in active mode during an alert. Panels a) and b) show the expected shaking (VI, strong), the countdown and the instructions on how to behave appearing on the screen during the event. Panel c) shows a screenshot of the screen once the earthquake is finished and the ground shaking has passed. Two buttons (green and red) appear on the screen to easily communicate the proper condition and position to a list of predefined contacts. The maps were created using OpenStreetMaps (© OpenStreetMap contributors 2019. Distributed under a Creative Commons BY-SA License.).

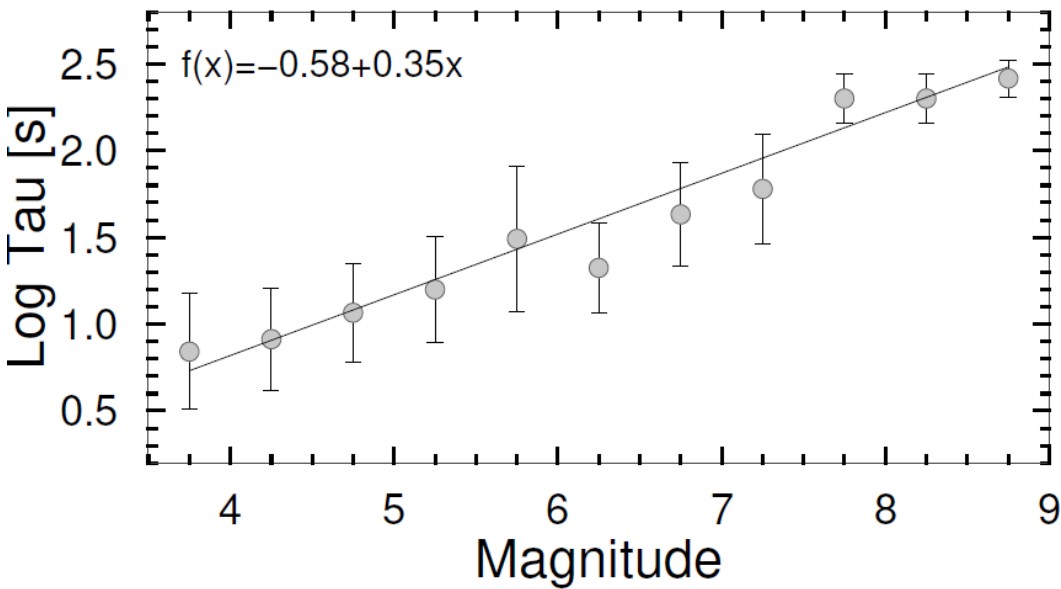

**Figure 7: Shaking duration vs. magnitude.** The figure shows the average shaking duration as a function of magnitude, in log-linear scale.

For each magnitude bin (0.5), the average value and its standard error are shown with light grey circles. The solid line is the best fit line, corresponding to the equation shown in the top-left corner of the plot.


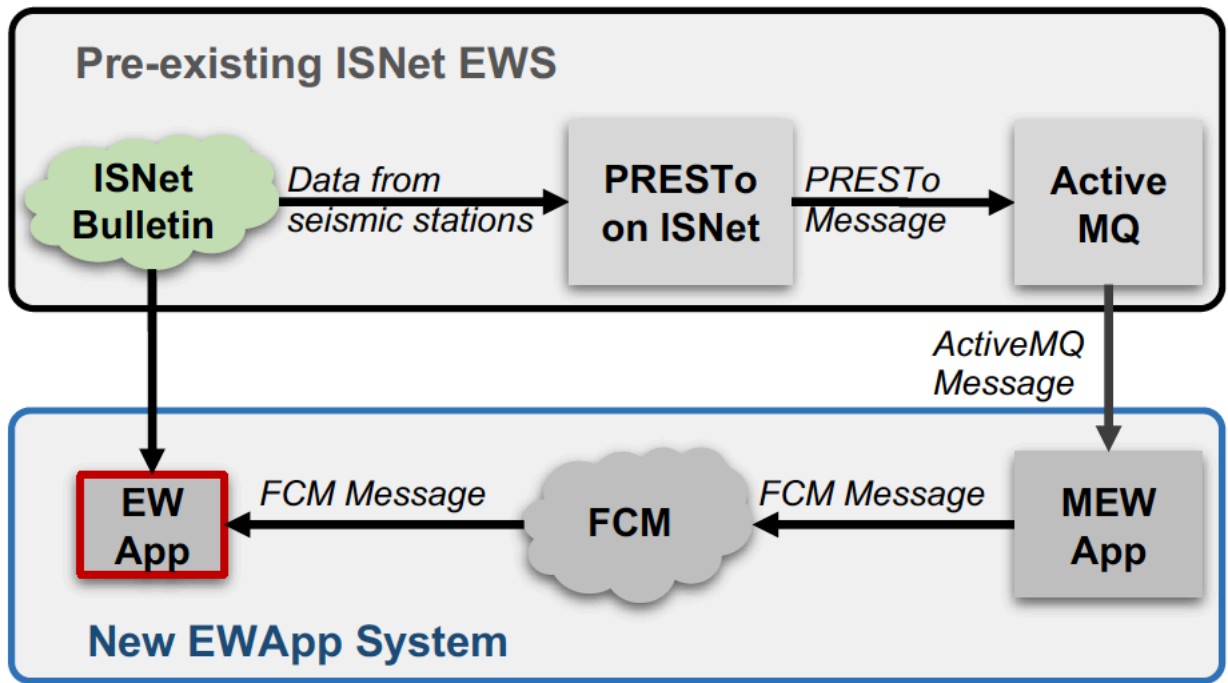

**Figure 8: Architecture of the EWApp.** The figure is a schematic representation of the whole architecture, involving both the network of stations, the EW software (ISNet network and PRESTo EW platform) and the EWapp. PRESTo processes the ISNet stations waveforms, sending evolutive earthquake information messages (in QuakeML format) to a message broker (ActiveMQ). The middleweare (MEWApp)
releases earthquake information to the FCM cloud, that broadcasts them to all the EWApp installations. The EWApp also downloads the revised ISNet bulletin on request.