# Peer review of "Design and implementation of a mobile device APP for network-based EEW systems: application to PRESTo EEWS in Southern Italy"

_Natural Hazards and Earth System Sciences, 2019_

## Referee Comment (RC1) · Anonymous Referee #1 · 18 Sep 2019

This short paper gives a good description of a newly developed android app to receive EEW messages from existing seismic network-based earthquake early warning system. During passive mode when there is no earthquake, the app serves as a viewer to display recent earthquake near the users. While in active mode, i.e. there is an earthquake detected by the EEW system and the user is close the earthquake, the app will receive the earthquake information message and decide whether to alert or not based on an empirical relationship. It also estimates the duration of the earthquake based on a relationship that extracted from the dataset the authors put together and

notify the end of the event to the users. After the event, the users can also report their conditions through the app to predefined contacts or social media account. Overall this paper is well written and easy to get the idea of the functionality of the app. Below are my comments/suggestions.

\* By design, the app will decide whether to alert to the user or not locally on the phone. This means that the system needs to send a message to all the phones within certain distances. This may not scale well if there are many users within the distance. Maybe add some sentences about why the authors decide this approach on the phone instead of computing on the server and only send to a subset of phones.

\* Line 186, 'As for the alert levels, these are progressively updated following the output in terms of location ...', does this also send to all the phones progressively? This sounds not efficient if there are a lot of phones.

\* Besides, if the estimated magnitude or the location of the event change (large enough), is that will also change the alert at the users' ends? For example, someone gets a warning due to the computed PGV exceeds the threshold, but a later update with the change of the location/magnitude, then this user's computed PGV value drop below the threshold, the alert cancel? or something else? Please specify in the paper.

\* The estimation of the end of the event (or duration of the event) is not convincing without distance. In the supplementary material, it is said several authors have shown that for regional distances, the dependency on distance can be negligible. The references are missing in the supplementary material, please add that.

\* Besides, the estimation of the end of the event may seem a little too complicated in real-time and prone to errors (may give wrong directions to users, especially for large earthquake with finite source). For example, based on this equation, an M7 earthquake will have about 75s duration, while if the estimation is M6.5, it will be 50s, which is 25s less. And this is all happening in real-time to the users, and they may use this information to take actions. This seems to contradictory to the findings in (Allen et al.

2018), where simpler and concise messages are preferred. Of course, this is not asking the authors to change this, because everyone has different design philosophy. But sometimes the perspectives from scientists are different from the public's views. I hope the authors can get some feedback from the public users instead of a few academic researchers.

\* For this app to work as expected, or a measure of the success of this app, are the delivery time latency and successful delivery rate, which are missing from this paper. Do you have any data about the latency from PRESTo send in the message to the time the phone acknowledge message received? And what's the successful delivery rate, i.e. what percentage of phones do get alert vs not (but supposedly should be). These are some of the key measurements for this type of apps, I hope the authors can add some of these numbers, otherwise, we won't know whether the app is really useful or not.

\* How many people are using this app right now? What's the plan for getting more people to use it? Since getting people to download the app is one thing, keeping them using the app is another thing. Do the authors have any plan to attract the public to use it?

\* If the user doesn't have location service on (likely for many of the users will do), they will not get a warning?

\* Are there any privacy issues of using user's location on phones? How do the authors handle these issues?

\* line 120, can you also label the broadbands in figure 1?

\* line 158, there is a typo 2-4

Ref: Allen R. et al 2018, Lessons from Mexico's Earthquake Early Warning System, EOS

СЗ

---

## Referee Comment (RC2) · Anonymous Referee #2 · 11 Oct 2019

This paper addresses a key component of an earthquake early-warning system – a mobile app for the notification of users of earthquake activity and alerts for potentially strong shaking at the user location. The paper gives a concise and apparently complete overview of the architecture, algorithms and operation of the PRESTo EEWS and of the ISNet EWApp, but lacks discussion of the expected performance of the app and alerts.

Specific comments:

1. The paper lacks a section on the expected performance of the EEWS and the EW

App with regards to alert timing and reliability. Presentation and analysis of the known, past performance of PRESTo EEWS for detecting and locating earthquakes for ISNet along with expected (and observed, if available) temporal latencies and uncertainties in the ISNet EWApp algorithms and infrastructure should allow the authors to estimate performance. The authors should discuss: alert timing (typical lead times before strong shaking, blind zones, ...; a map would be useful), reliability (error in alert shaking levels relative to possible real shaking levels, ...) for users in different parts of the ISNet region, and issues of latency and delivery reliability raised by reviewer #1.

2. It may be better to issue a generic message like "[intense/strong/weak/..] earthquake shaking expected [in N seconds]" instead of "Drop! Cover! Hold On!" The latter is appropriate for wood-frame structures and probably (statistically) appropriate for denser areas of masonry structures, but may be bad advice for specific cases such as low, masonry and stone construction in rural areas. What happens legally if, though a vast majority of people were safer following the "Drop! Cover! Hold On!" action, some are injured or killed because they followed this advice and remained in structures which collapsed, though they could have easily exited to an open space? Information on actions to take might be disseminated and discussed before hand independently of the app (public information channels, schools, work places) so that people learn that actions to take in strong earthquake shaking are related to their situation. Even if it is decided that "Drop! Cover! Hold On!" is the best and only action to promote in the ISNet EWApp user region, is it appropriate to use this unique action message in the app?

Technical corrections:

1. The phrases "distant event" and "closer event" and similar are used often but sometimes are ambiguous whether the distance is from the user or from ISNet.

2. Line 116: "them" → "themselves"

3. Line 119: "components" → "component"

4. Line 120: " the well recording of teleseismic events" → "high-quality recording of teleseismic events"

5. Line 129: " the damaging" → "damaging"

6. Line 144: "smaller than" → "less than"

7. Line 157: " 2 / 4" → "2-4"

8. Lines 167 and 247: Should the term "issue the warning" be "issue an alert"?

9. Line 168: "a faraway earthquake" → "an earthquake far from the user"

10. Line 188: "In the event of a damaging earthquake" - might be better phrased as "In the event of strong shaking and a potentially damaging earthquake". But is this level simply IMM pred ≥ IMM*? In this case, does not this smartphone display mode apply to all events described in this paragraph?

11. Line 205: "is pretty insensitive to distance" → "has weak dependence on distance"

12. Line 208: "has the possibility to notify his own health condition, and to communicate it to a list of pre-defined contacts" → "has the possibility to communicate their own condition to a list of pre-defined contacts"

13. Line 217: "estimate" → "estimates"

14. Line 222: "configured to not" → "not configured to"

15. Line 226: "The alerts sent via FCM can in fact awaken the device even when it is in standby" - is this done through the phone SIM card, mobile phone reception, or Internet? Also remove "in fact".

16. Line 239: "perform earthquake alert notifications, in the event of a relevant damaging earthquake." → "send earthquake notifications, including alerts for expected strong shaking and a potentially damaging earthquake at the user location."

17. Line 250: "expected damage" → "expected damage potential"

18. Line 251: "as well as the instructions for the behavior during emergency times." → "as well as instructions for mitigating actions for the user."

19. Line 254: "double communication way" → "two-way flow of information to and from the user."

20. Line 256: "at the conclusion of the emergency time" → "after the alert period (expected shaking duration) has ended"

21. Figure 5: The statement "Earthquake intensity Not Felt in Sant'Angelo del Lombardi" reads as if the earthquake is not felt in Sant'Angelo del Lombardi, instead of an earthquake in Sant'Angelo del Lombardi is not felt at the user location. Would be better something like: "Earthquake in Sant'Angelo del Lombardi: Your expected intensity: Not Felt"

22. Figure 8: Label which path is an "alert" or "push notification" and which is the "passive mode".

---

## Author Comment (AC1) · 26 Nov 2019

The comment was uploaded in the form of a supplement: https://www.nat-hazards-earth-syst-sci-discuss.net/nhess-2019-266/nhess-2019-266-AC1-supplement.pdf

---

## Author Response (AR1)

**Design and implementation of a mobile device APP for network-based EEW systems: application to PRESTo EEWS in Southern Italy**

By Simona Colombelli, Francesco Carotenuto, Luca Elia and Aldo Zollo

We thank the two Reviewers for their comments and feedback that significantly helped us to improve the original manuscript. We have responded to each comment separately and modified the manuscript, where necessary. We hope that the current version of the manuscript satisfies the two Referees as well as the Editor.

**Anonymous Referee #1**

This short paper gives a good description of a newly developed android app to receive EEW messages from existing seismic network-based earthquake early warning system. During passive mode when there is no earthquake, the app serves as a viewer to display recent earthquake near the users. While in active mode, i.e. there is an earthquake detected by the EEW system and the user is close the earthquake, the app will receive the earthquake information message and decide whether to alert or not based on an empirical relationship. It also estimates the duration of the earthquake based on a relationship that extracted from the dataset the authors put together and notify the end of the event to the users. After the event, the users can also report their conditions through the app to predefined contacts or social media account. Overall this paper is well written and easy to get the idea of the functionality of the app. Below are my comments/suggestions.

1) By design, the app will decide whether to alert to the user or not locally on the phone. This means that the system needs to send a message to all the phones within certain distances. This may not scale well if there are many users within the distance. Maybe add some sentences about why the authors decide this approach on the phone instead of computing on the server and only send to a subset of phones.

>> The reviewer is right, we let the app decide whether to alert the user or not, directly on the phone. This has been decided in order to have an independent tool to compute the predicted ground shaking, that does not require the intensity as an input parameter. The EWApp can be, in principle, interfaced to any EEW platform providing basic source parameters (i.e., location and magnitude) of an ongoing event. This is the same principle of other similar tools, such as the User Display (Böse et al., 2014) and the Earthquake Early Warning Display (Cauzzi et al., 2016), which also compute the expected MMI intensity and the remaining warning time at the user position, based on the information provided by the network-based EEWS.

Our system sends alert messages to all phones on which the app is installed, with no filters based on the distance. Then, depending on the relative position between the earthquake source and the user, a different message is displayed on the app (passive mode; active mode with no alert; active mode with alert). To handle the issue of a large number of users, we implemented a Fire Cloud Messaging (FCM) server for managing the alert delivery, so that no big hardware and software infrastructures are necessary. We included a sentence in the revised version of the manuscript to justify our approach.

2) Line 186, 'As for the alert levels, these are progressively updated following the output in terms of location ...', does this also send to all the phones progressively? This sounds not efficient if there are a lot of phones.

>> This comment is related to the following one. Please, see our response below.

3) Besides, if the estimated magnitude or the location of the event change (large enough), is that will also change the alert at the users' ends? For example, someone gets a warning due to the computed PGV exceeds the threshold, but a later update with the change of the location/magnitude, then this user's computed PGV value drop below the threshold, the alert cancel? or something else? Please specify in the paper.

>> We apologize for not explaining this concept clearly in the manuscript. The alert levels are progressively updated every second, based on the estimated outputs of location and magnitude of the event from the EW system. To avoid discontinuities and jumps in the displayed parameters, due to small changes in the earthquake location, the alert message is updated only when the estimated intensity changes (both increasing or decreasing are possible). In this case, the countdown is also updated when the real-time, current lead-time value differs from the previous one for more than 5 seconds. As for the alert cancellation, following the same strategy adopted by PRESTo (Satriano et al., 2010), the app does not include the possibility of cancelling the alert, once it has been released. In case of an expected intensity starting high and then dropping below the threshold, the user will keep on seeing the alert message on the display, but with a smaller estimated value of intensity. We better explained the updates of alerts in the revised version of the manuscript.

4) The estimation of the end of the event (or duration of the event) is not convincing without distance. In the supplementary material, it is said several authors have shown that for regional distances, the dependency on distance can be negligible. The references are missing in the supplementary material, please add that.

>> We understand the Reviewer skepticism, but we preliminary investigated the dependency of the duration on distance, as reported in the Supplementary Material. For the analyzed data, covering a distance range from 0 to 200 km, we found that the duration of the shaking mainly depends on the earthquake magnitude and has no significant dependency on the source-to-receiver distance (see Figure S2). The same assumption is also used by other authors when computing the duration magnitude for local and regional distances (Bindi et al., 2005; Castello et al., 2007; Del Pezzo et al., 2003; Real and Teng, 1973). We included the missing references in the text.

5) Besides, the estimation of the end of the event may seem a little too complicated in real-time and prone to errors (may give wrong directions to users, especially for large earthquake with finite source). For example, based on this equation, an M7 earthquake will have about 75s duration, while if the estimation is M6.5, it will be 50s, which is 25s less. And this is all happening in real-time to the users, and they may use this information to take actions. This seems to contradictory to the findings in (Allen et al. 2018), where simpler and concise messages are preferred. Of course, this is not asking the authors to change this, because everyone has different design philosophy. But sometimes the perspectives from scientists are different from the public's views. I hope the authors can get some feedback from the public users instead of a few academic researchers.

>> We thank the Reviewer for this interesting feedback. In the current version of the app, the duration of the event is only used to inform the user that the strongest shaking has finished. At the end of the event, a different screen is shown on the display of the app (see Figure 6c of the main text), together with a specific voice message informing the user that the shaking has passed. We do not provide the expected duration to the user in real-time. The app indeed only displays the countdown from the current time to the arrival of the strongest shaking. Therefore, there is no risk of confusing the users about the actions to take, if this was the Reviewer's concern. Indeed, we agree with the Reviewer that simple and concise messages are preferred when communicating alerts to the public in real-time.

As for the user's feedback, we already included in our app the possibility for the users to send free feedback to the developers. We are now planning to prepare a dedicated questionnaire and ask the users to fill it. In the questionnaire, specific questions about the alert communication will be included, so that we will collect feedback directly from our users and will modify the app according to their suggestions. As for the public involvement, please see our response to comment #7. We included a sentence in the manuscript.

6) For this app to work as expected, or a measure of the success of this app, are the delivery time latency and successful delivery rate, which are missing from this paper. Do you have any data about the latency from PRESTo send in the message to the time the phone acknowledge message received? And what's the successful delivery rate, i.e. what percentage of phones do get alert vs not (but supposedly should be). These are some of the key measurements for this type of apps, I hope the authors can add some of these numbers, otherwise, we won't know whether the app is really useful or not.

>> We apologize for missing such key measurements. We included both here and in the revised version of the manuscript a new paragraph dedicated to the performance in terms of latency and delivery rate.

We carried out a set of tests to quantify the performance of the EWApp in terms of latency times and successful delivery rate. Specifically, we measure the latency between the time of the alert as sent by the Middleware App and the calculation of the expected intensity and lead time by the EWApp, as explained below.

For each message $i$ sent by the FCM and received by the smartphone, the total latency (*Delay*) introduced by the app can be obtained as the difference between the time of the reception of the message ($T_{IN\_i}$) and the time of the output parameter release ($T_{OUT\_i}$):

$$Delay_{\_i} = T_{OUT\_i} - T_{IN\_i}$$

where $T_{IN\_i}$ is the time at which the message is sent by the FCM to the smartphone, $T_{OUT}$ is the time at which the output parameters computed by the EWApp are available and released and i represents each available measurement.

By definition, the total latency is positive and contains the computational latency due to the smartphone operations, the time required to call the web service and the delay between the sending of the message by the middleware server and the actual sending by FCM to the smartphone.

For the testing phase, we used a sample of 9 Android smartphones, with heterogeneous technical and usage characteristics. We collected data by automatically sending test alerts to all the available users for 15 days, including both working days and weekends. As a testing earthquake, we used the Mw 6.9 1980 Irpinia earthquake, which is the strongest event occurred during the last four decades in the region of interest and represents the target event of our EWS. The playback test consisted in sending 36 messages in the form of a random swarm (1 to 4 events every 30 to 40 sec, randomly created). Each swarm is repeated with a random period between 6 and 9 min, for a total number of 38808 alert events. For each available alert, we measure the total latency, as defined above. Figure 1 shows the percentage histograms of the total latency. We characterize the unimodal, non-symmetric distribution with its mode value, equal to **1.034s**, and representing our best estimates of the latency introduced by the EWapp.

[Figure]

**Figure 1. Performance of the EWapp in terms of total latency**. The figure shows the percentage histograms of the total latency. The distribution is unimodal and non-symmetric, with a mode value equal to 1.034s.

Following a similar logic that has been used for the latency computation, we estimated the delivery rate as the difference between the number of received messages and the number of output messages as released by the smartphones. In this case, the test was conducted in a different way, to avoid any potential crowding of the FCM server, which could produce frequent lost messages, resulting in a non-realistic estimate of the delivery rate. We used the same sample of 9 Android smartphones, ensuring that all the smartphones were turned on during the experimentation, and sent a total number of about 1200 alert simulations. Single smartphone delivery rate values range from 41% to 98%, with an average value of **80%**.

It is worth to note that, both for the latency time and for the delivery rate, the observed performance is strongly affected by the use of the standard FCM cloud. Compared to a custom, dedicated solution, this has the advantage of being free, supported by default and tightly integrated in the Android platform, for instance it allows message delivery even when the smartphone is under sleep. On the other hand, we cannot control or improve the policies and limits on message dissemination by the service.

7) How many people are using this app right now? What's the plan for getting more people to use it? Since getting people to download the app is one thing, keeping them using the app is another thing. Do the authors have any plan to attract the public to use it?

>> The app has been currently released to a limited number of academic users (about 30), which are testing the tool and providing confidential feedback to the developers. The app has been originally developed to be the front-end of PRESTo EEWS, as an additional way of disseminating the alerts released by the system to the users in the area of interest of the regional network, where the ISNet monitoring infrastructure is deployed. PRESTo is currently running in an experimental phase and a limited number of selected users is receiving alerts through sms and emails. For the EWApp, we foresee a similar prototypal experimentation, with an initial involvement of a restricted number of specialized users, that will provide feedback to correct potential malfunctions of the tool. Then, within the framework of National/European projects and initiatives related to the EWS, we can gradually increase the number of users, including public stakeholders and citizens. For example, some schools in the Campania-Lucania region where the EWS is running could be selected as initial experimenters where to test a wider release of the app. Given the very specific target users of our app, we believe that once it has been downloaded and installed, the risk of being cancelled is rather low, also because the app can work in background mode and when the energy save mode is activated, so that no specific action is required by the user to keep it operative. We included a comment in the revised version of the manuscript to discuss the plan for releasing the app.

8) If the user doesn't have location service on (likely for many of the users will do), they will not get a warning?
>> At the first installation of the app, the user can select among two options. The first one is to use a fixed position, while the second one is to activate the location function, in order to have continuously updated position measurements, for a reliable computation of the distance from the source, and therefore, of the expected shaking. If the location service is not enabled, the app will use the last available position of the smartphone and the users will receive the warning anyway. We specify this in the revised version of the manuscript.

9) Are there any privacy issues of using user's location on phones? How do the authors handle these issues?
>> Yes, the user's location is hidden to the developers, to guarantee privacy policies. We specify this in the revised version of the manuscript.

10) line 120, can you also label the broadbands in figure 1?
>> We apologize for not accounting for this request. When reviewing the manuscript, we realized that the sentence on line 120 is redundant, and we therefore eliminated the comment about broadband stations. Our Early Warning System PRESTo, indeed, does not use broadband stations, but it only uses strong motion accelerometers.

11) line 158, there is a typo 2-4
>> Thanks, we corrected the typo.

Ref: Allen R. et al 2018, Lessons from Mexico's Earthquake Early Warning System, EOS

**Anonymous Referee #2**

This paper addresses a key component of an earthquake early-warning system – a mobile app for the notification of users of earthquake activity and alerts for potentially strong shaking at the user location. The paper gives a concise and apparently complete overview of the architecture, algorithms and operation of the PRESTo EEWS and of the ISNet EWApp, but lacks discussion of the expected performance of the app and alerts. Specific comments:

12) The paper lacks a section on the expected performance of the EEWS and the EW App with regards to alert timing and reliability. Presentation and analysis of the known, past performance of PRESTo EEWS for detecting and locating earthquakes for ISNet along with expected (and observed, if available) temporal latencies and uncertainties in the ISNet EWApp algorithms and infrastructure should allow the authors to estimate performance. The authors should discuss: alert timing (typical lead times before strong shaking, blind zones, ...; a map would be useful), reliability (error in alert shaking levels relative to possible real shaking levels, ...) for users in different parts of the ISNet region, and issues of latency and delivery reliability raised by reviewer #1.
>> We agree with the Reviewer that the performances of the app were not sufficiently described in the original manuscript. This comment is very similar to comment #6 raised by Reviewer 1, and we therefore prepared a unique response, accounting for the points raised by the two Reviewers. Please, see our response above.

However, a relevant aspect needs to be clarified. The EWapp that we developed has been conceived to be interfaced with any regional, network-based EW platform (such as PRESTo, VS, Elarms) and in the present manuscript, we focus on the use of the app to broadcast the alerts issued by PRESTo, which is currently operative in Southern Italy. The purpose of the paper is to present the EWApp and its main functions, both in the absence and in presence of earthquakes. The aim of this paper is not to present PRESTo EWS and its warning capabilities (including lead-times, blind zones and ground shaking prediction errors within ISNet region). These have been already presented and largely discussed in the reference paper (Satriano et al., 2010). Instead, we included here and in the revised version of the manuscript a dedicated section where we discuss the performance of the EWApp, in terms of latency times and successful rate of alert transmission. See our response to comment #6 raised by Reviewer 1. We hope that the new section satisfies the request of Reviewer #2 as well.

To provide some numbers on how fast can the ISNet EW system be in releasing an alert, we analyzed the real-time performance of PRESTo. Over the last 3 years (2016-12-09 to 2019-11-18) PRESTo declared 74 earthquakes. Figure 2 reports the statistics of the seconds passed from the first P-wave arrival at the ISNet network (first measurable impact of the earthquake at the ground) to the first alert released by PRESTo. We see that 34 % of the alerts were released within 4 seconds, 62% within 5 seconds, 84% within 6 seconds and 96% within 7 seconds. Note that these times include all the delays introduced by acquisition, telemetry and processing, plus eventual seismic network issues (i.e. some stations not working). Furthermore, PRESTo is configured to wait for the arrival of P-waves at a minimum of 5 stations before declaring an event, in order to avoid false alerts.

[Figure]

**Figure 2. Distribution of the first PRESTo alert time for the ISNet network, over the last three years.** The histograms report the seconds required from the first P-wave arrival at the first station to the release of the first alert (i.e. first location and magnitude) by PRESTo.

13) It may be better to issue a generic message like "[intense/strong/weak/..] earthquake shaking expected [in N seconds]" instead of "Drop! Cover! Hold On!" The latter is appropriate for wood-frame structures and probably (statistically) appropriate for denser areas of masonry structures, but may be bad advice for specific cases such as low, masonry and stone construction in rural areas. What happens legally if, though a vast majority of people were safer following the "Drop! Cover! Hold On!" action, some are injured or killed because they followed this advice and remained in structures which collapsed, though they could have easily exited to an open space? Information on actions to take might be disseminated and discussed before hand independently of the app (public information channels, schools, work places) so that people learn that actions to take in strong earthquake shaking are related to their situation. Even if it is decided that "Drop! Cover! Hold On!" is the best and only action to promote in the ISNet EWApp user region, is it appropriate to use this unique action message in the app?

>> We agree with the Reviewer that the message "Drop! Cover! Hold On!" might not be appropriate for a generic user location/condition, but we believe there has been a misunderstanding, due to our incomplete explanation. We clarify this issue both here and in the revised version of the manuscript.

In case of alert, the app shows two essential pieces of information, which are the countdown (available lead-time at the user's position) and the expected intensity. This last is given both in roman numbers and in the form of a text description, such as "intense/strong/weak", following the intensity scale definition of Faenza & Michelini, 2010). The countdown and the expected intensity are shown on the display and are also announced by the voice message. Then, the last five seconds of warning, an additional voice message saying "Save yourself" is given. Such a communication format provides the most relevant pieces of information (shaking and time) both in the form of text and in the form of sound, thus allowing any user in a generic condition/location to properly react, with no distinction between indoor/outdoor positions.

For the entire duration of the alert, the icon picture showing the "Drop! Cover! Hold on" behavior is also shown at the bottom of the screen, but no related instructions are given by the voice message. This is a standardized alert message, suitable for indoor EW applications. As already explained in comment #7 by Reviewer 1, we foresee for the app a prototypal experimentation, with an initial involvement of a restricted number of specialized users, that will provide feedback to the developers. Then, within the framework of National/European projects and initiatives related to the EWS, we will gradually increase the number of users, including public stakeholders and citizens. By this time, we will involve a dedicated team of expert social scientists to find out the optimal way for communicating the alerts, and to modify the output format of the current version of the EWapp. We included a comment in the revised version of the manuscript.

Technical corrections:

1. The phrases "distant event" and "closer event" and similar are used often but sometimes are ambiguous whether the distance is from the user or from ISNet.
>> OK, we clarified it.
2. Line 116: "them" ! "themselves"
>> We apologize for not including this suggestion, but we believe the word "them" being more appropriate.
3. Line 119: "components" ! "component"
>> OK, done
4. Line 120: " the well recording of teleseismic events" ! "high-quality recording of teleseismic events"
>> OK, done
5. Line 129: " the damaging" ! "damaging"
>> OK, done
6. Line 144: "smaller than" ! "less than"
>> we changed "at a distance smaller than 200km" into "within 200km"
7. Line 157: " 2 / 4" ! "2-4"
>> OK, done
8. Lines 167 and 247: Should the term "issue the warning" be "issue an alert"?
>> OK, done
9. Line 168: "a faraway earthquake" ! "an earthquake far from the user"
>> OK, done
10. Line 188: "In the event of a damaging earthquake" - might be better phrased as "In the event of strong shaking and a potentially damaging earthquake". But is this level simply IMM pred _ IMM*? In this case, does not this smartphone display mode apply to all events described in this paragraph?
>> Yes, this smartphone display mode applies to all events described in this paragraph. We changed the original text "In the event of a damaging earthquake", to "During the alert".

11. Line 205: "is pretty insensitive to distance" ! "has weak dependence on distance"
>> We changed this sentence an included an additional comment.
12. Line 208: "has the possibility to notify his own health condition, and to communicate it to a list of pre-defined contacts" ! "has the possibility to communicate their own condition to a list of pre-defined contacts"
>> We changed "his own health condition" into "the proper condition"
13. Line 217: "estimate" ! "estimates"
>> OK, done
14. Line 222: "configured to not" ! "not configured to"
>> OK, done

15. Line 226: "The alerts sent via FCM can in fact awaken the device even when it is in standby" - is this done through the phone SIM card, mobile phone reception, or Internet? Also remove "in fact".
>> The alert are sent through the FCM service, which requires internet connection. We specify this in the revised version of the manuscript.

16. Line 239: "perform earthquake alert notifications, in the event of a relevant damaging earthquake." ! "send earthquake notifications, including alerts for expected strong shaking and a potentially damaging earthquake at the user location."
>> OK, done
17. Line 250: "expected damage" ! "expected damage potential"
>> OK, done
18. Line 251: "as well as the instructions for the behavior during emergency times." ! "as well as instructions for mitigating actions for the user."
>> OK, done
19. Line 254: "double communication way" ! "two-way flow of information to and from the user."
>> OK, done
20. Line 256: "at the conclusion of the emergency time" ! "after the alert period (expected shaking duration) has ended"
>> OK, done

21. Figure 5: The statement "Earthquake intensity Not Felt in Sant'Angelo del Lombardi" reads as if the earthquake is not felt in Sant'Angelo del Lombardi, instead of an earthquake in Sant'Angelo del Lombardi is not felt at the user location. Would be better something like: "Earthquake in Sant'Angelo del Lombardi: Your expected intensity: Not Felt"
>> OK, done

22. Figure 8: Label which path is an "alert" or "push notification" and which is the "passive mode".
>> OK, done

**References**
Bindi, D., D. Spallarossa, C. Eva, and M. Cattaneo (2005). Local and duration magnitudes in the northwestern Italy, and seismic moment versus magnitude relationship, Bull. Seismol. Soc. Am. 95, 592–604.
Böse M. et al. (2014) CISN ShakeAlert: An Earthquake Early Warning Demonstration System for California. In: Wenzel F., Zschau J. (eds) Early Warning for Geological Disasters. Advanced Technologies in Earth Sciences. Springer, Berlin, Heidelberg
Castello, B., M. Olivieri, and G. Selvaggi (2007). Local and duration magnitude determination for the Italian earthquake catalogue, Bull. Seismol. Soc. Am. 97, 128–139.
Cauzzi C., Behr Y., Clinton J., Kästli P., Elia L., and Zollo A.; An Open-Source Earthquake Early Warning Display. Seismological Research Letters; 87 (3): 737–742. doi: https://doi.org/10.1785/0220150284
Del Pezzo, E., F. Bianco, and G. Saccorotti (2003). Duration magnitude uncertainty due to seismic noise: Inferences on the temporal pattern of G–R b-value at Mt. Vesuvius, Italy, Bull. Seismol. Soc. Am. 93, 1847–1853.
Satriano C., Elia L., Martino C., Lancieri M., Zollo A. and G. Iannaccone. PRESTo, the earthquake early warning system for Southern Italy: Concepts, capabilities and future perspectives. Soil Dyn Earthquake Eng (2011), 31 (2), 137-153 doi:10.1016/j.soildyn.2010.06.008